# Involvement of Gut Microbial Metabolites Derived from Diet on Host Energy Homeostasis

**DOI:** 10.3390/ijms23105562

**Published:** 2022-05-16

**Authors:** Akari Nishida, Yuna Ando, Ikuo Kimura, Junki Miyamoto

**Affiliations:** 1Laboratory of Molecular Neurobiology, Graduate School of Pharmaceutical Sciences, Kyoto University, Yoshidakonoe-cho, Sakyo-ku, Kyoto 606-8501, Japan; nishida.akari.75f@st.kyoto-u.ac.jp (A.N.); kimura.ikuo.7x@kyoto-u.ac.jp (I.K.); 2Laboratory of Molecular Neurobiology, Graduate School of Biostudies, Kyoto University, Yoshidakonoe-cho, Sakyo-ku, Kyoto 606-8501, Japan; ando.yuna.83n@st.kyoto-u.ac.jp; 3Department of Applied Biological Science, Graduate School of Agriculture, Tokyo University of Agriculture and Technology, Fuchu-shi, Tokyo 183-8509, Japan

**Keywords:** gut microbiota, diet, metabolite, obesity, energy homeostasis

## Abstract

Due to the excess energy intake, which is a result of a high fat and high carbohydrate diet, dysfunction of energy balance leads to metabolic disorders such as obesity and type II diabetes mellitus (T2DM). Since obesity can be a risk factor for various diseases, including T2DM, hypertension, hyperlipidemia, and metabolic syndrome, novel prevention and treatment are expected. Moreover, host diseases linked to metabolic disorders are associated with changes in gut microbiota profile. Gut microbiota is affected by diet, and nutrients are used as substrates by gut microbiota for produced metabolites, such as short-chain and long-chain fatty acids, that may modulate host energy homeostasis. These free fatty acids are not only essential energy sources but also signaling molecules via G-protein coupled receptors (GPCRs). Some GPCRs are critical for metabolic functions, such as hormone secretion and immune function in various types of cells and tissues and contribute to energy homeostasis. The current studies have shown that GPCRs for gut microbial metabolites improved host energy homeostasis and systemic metabolic disorders. Here, we will review the association between diet, gut microbiota, and host energy homeostasis.

## 1. Introduction

In recent decades, it has become a widely accepted fact that the gut microbiota performs multiple functions and influences host metabolism and health. Most gut bacteria belong to five phyla, with approximately 80–90% belonging to Firmicutes and Bacteroidetes and the remaining to Actinobacteria, Proteobacteria, and Verrucomicrobia [1]. There is every reason to believe that the diversity and composition of the gut microbiota can influence not only its own functions but also the metabolism and health of the host. In particular, among many factors that could influence the composition of any gut microbiota, the diet of the host has received the most attention because it presents the simplest approach for administrating potential preventive or therapeutic interventions against various metabolic disorders [2]. Aside from affecting host homeostasis, dietary components can interact with the gut microbiota and change the luminal environment in the gut. Recent advances in omics analysis for characterizing the gut microbial environment have accelerated our understanding of how diets modulate the gut microbiota and, most importantly, affect host homeostasis [3]. Changes in dietary patterns lead to rapid and reversible population changes in the dominant gut microbes [4]. The meta-analyses have examined the differences in the gut microbiota of obese and lean individuals, finding that Roseburia and Mogibacterium are significantly enriched in obese individuals, while Anaerovorax, Oscillibacter, Pseudoflavonifractor, and Clostridium IV are depleted [5]. In fact, the extent of the change of gut microbial composition is highly variable across studies and seems to be highly dependent on potential confounding factors, especially the diet. Metabolic disorders, such as obesity and type II diabetes mellitus (T2DM), have been described as the outcome of complex crosstalk between individual genotypes, aging, environmental factors, dietary pattern, and the gut microbiota [6]. The nutrition of individuals seems to be the most important element that shapes the gut microbiome [7]. Conversely, the characteristics of gut microbiota may influence the response to dietary intervention. Crosstalk between gut microbiota and diet may partly explain why microbiota alterations are associated with an obese phenotype in observational studies in humans and can also explain the interindividual response to dieting in the specific context of obesity [8,9]. For those reasons, we will first realize how energy intake, dietary habits, and also the ingestion of specific nutrients, may influence the composition and activity of gut microbiota, thereby having an influence on host metabolism.

Free fatty acids (FFAs) in dietary lipids are not only an essential energy source but also function as signaling molecules through G-protein coupled receptors (GPCRs), which regulate various cellular processes and physiological functions [10]. FFAs are classified by carbon chain length, where short-chain fatty acids (SCFAs) have no more than 6 carbon atoms, medium-chain fatty acids have from 6–12 carbon atoms, and long-chain fatty acids (LCFAs) have greater than 12 carbon atoms. SCFAs, (e.g., acetate, propionate, and butyrate) are produced by the gut microbial fermentation of dietary fibers and can activate GPR41 and GPR43 [11]. By contrast, essential fatty acids, such as linoleic acid (ω6, C18:2) and α-linolenic acid (ω3, C18:3), must be obtained from the diet because humans cannot synthesize the ω3 or ω6 double bonds. Because these LCFAs are recognized by GPR40 and GPR120, these activated GPCRs play a pivotal role in energy homeostasis [10].

In this review, we describe the interactions among the gut microbiota, diet, and the host. We also discuss current knowledge on how diets affect the gut microbiota and how gut microbial metabolites influence the physiology and health of the host.

## 2. Gut Microbiota and Obesity

### 2.1. Gut Microbiota

The gut microbiota is involved in energy harvesting and storage as well as in the fermentation and absorption of indigestible carbohydrates. It is well known that gut microbiota dysbiosis is correlated with metabolic disorders, such as obesity and T2DM [12,13]. Not only the gut microbiota composition has been reported to exhibit strong pathological and physiological associations with metabolic disorders, but the ratio of Firmicutes to Bacteroidetes has also been found to be crucial for maintaining normal gut homeostasis. In the human study, the ratio of Firmicutes to Bacteroidetes has been found to increase with the increase in the body mass index. Inoculation of the gut microbiota from twins (discordant for obesity) into mice results in the induction of obesity, indicating a causal role of the gut microbiota in the pathogenesis of metabolic disorders [14]. Additionally, the transplantation of fecal microbiota from healthy individuals to patients with T2DM enhanced insulin sensitivity in part of the study cohort. The efficacy of the fecal microbiota transplantation approach clearly depended on the composition of the recipient gut microbiota [15].

*Akkermansia muciniphila*, as a member of Verrucomicrobia in gut microbiota, can use mucin as its sole carbon and nitrogen source. Many studies have shown that the abundance of *A. muciniphila* in the gut is correlated with metabolic disorders [16,17]. Moreover, as demonstrated in humans and different mouse models, *A. muciniphila* could play a crucial role in modulating the obesity parameters, such as reducing the body weight, fat mass, hip circumference, caloric intake, mesenteric fat weight, subcutaneous fat weight, epididymal fat weight, total fats, and energy efficiency [18]. These findings in metabolic disorders suggest that *A. muciniphila* in the gut microbiota directly regulates energy homeostasis and glucose tolerance in the host (Table 1).

Given that diet is a major factor influencing the gut microbiota, variations in the diet would be a key aspect contributing to the diversity and composition of the gut microbes. David et al. reported that volunteers on either a vegetable-based or an animal-based diet for five consecutive days, whereupon severe temporal alterations in the composition of the gut microbiota were induced. The animal-based diet increased the abundance of bile-tolerant microbiota (*Alistipes*, *Bilophila*, and *Bacteroides*) and decreased the levels of Firmicutes that fiber-fermenting microbiota (*Roseburia*, *Eubacterium rectale*, and *Ruminococcus bromii*) [7]. Additional studies have demonstrated the composition of the gut microbiota is affected by subtle differences in dietary lipids [43,44], fibers [45,46], and food additives [47,48].

### 2.2. Gut Microbial Metabolites

Currently, growing evidence reveals that the interactions between the wide range of gut microbial metabolites derived from diets and host metabolism are essentially but differentially involved in the development of metabolic disorders, with diverse mechanisms being proposed to account for the related metabolic alternations. For example, the gut microbiota converts dietary phosphatidylcholine (L-carnitine) to trimethylamine (TMA), a pro-atherosclerotic metabolite. TMA is rapidly further oxidized by hepatic enzymes to form trimethylamine N-oxide (TMAO), which acts as a trigger for the pathogenesis of cardiovascular disease [21,22]. Other metabolites that are produced upon the gut microbial metabolism of amino acids may be of interest for the management of metabolic disorders. The mechanisms by which branched-chain amino acids (BCAAs) induce insulin resistance are complex and controversial, and the reasons for the increase in BCAAs in patients with obesity and insulin resistance are equally unknown [19]. However, BCAAs, (e.g., leucine, isoleucine, and valine) have been proposed to be harmful metabolites, given that serum metabolomics has shown their levels to be characteristically high in patients with insulin resistance. Additionally, driver species such as *Prevotella copri* and *Bacteroides vulgatus* can induce both insulin resistance and an increase in serum BCAA levels [49]. Recently, it has been found that imidazole propionate, a gut microbial metabolite derived from dietary histidine, is present at higher concentrations in patients with T2DM. The administration of imidazole propionate to mice exacerbated their glucose tolerance and impaired insulin signaling through the activation of the p38γ/p62/mTORC1 pathway, suggesting that this microbial metabolite may contribute to the pathogenesis of T2DM [23,24]. These studies confirm that the crosstalk between dietary components and the gut microbiota may influence host homeostasis and lead to the regulation of gut microbial metabolites that may have beneficial or harmful effects on the host (Figure 1) [50]. These results expect that this area of research should open up new therapeutic avenues for metabolic disorders and their comorbidities. 

Bile acids are formed by the microbiota from host cholesterol and are another group of metabolites with a profound effect on human health [51]. The secondary bile acids are generated by gut microbiota from primary bile acids in the lower part of the small intestine and the colon. They were originally thought only to act as soaps that solubilize dietary fats to promote their absorption, but over the past two decades, it has become clear that they act as signaling molecules and bind to distinct receptors such as G-protein coupled bile acid receptor 1 (also known as TGR5) and the bile acid receptor FXR (Farnesoid X receptor) [52]. Both TGR5 and FXR have a major impact on host metabolism and, therefore, an altered microbiota might affect host homeostasis by modulating the signals passing through these receptors [52]. The ability to metabolize tauro-β-muricholic acid, a natural FXR antagonist, is essential for the microbiota to induce obesity and steatosis, as well as reduced glucose and insulin tolerance [25,26,27,28]. These results through two receptors are mediated by altered gut microbiota and bile acid metabolism. Bariatric surgery is associated with an altered microbiota and bile acid metabolism [53,54]. Mechanistic links between bile acids and bariatric surgery show that functional FXR signaling is required for reduced body weight and improved glucose tolerance following vertical gastrectomy [54]. Similarly, TGR5 is necessary for the improvement of glucose metabolism according to this procedure. Germ-free mice that received a fecal transplant from people who had undergone bariatric surgery (Roux-en-Y gastric bypass) 10 years earlier gained less fat than did mice that were colonized by microbiota from obese people [53]. Some of the beneficial effects of bariatric surgery could be mediated by an altered microbial metabolism of bile acids, which affects their signaling capacity. Other mechanisms and metabolites could also play important roles.

Collectively, these data suggest that the gut microbiota may mediate or even orchestrate events locally in the gut through the influence of diet, which can alter their metabolite signaling for regulating systemic metabolism. Although further research will be needed to elucidate ways to leverage gut microbiota in the prevention and treatment of obesity, this signaling interaction with the host may promote the development of potentially relevant therapeutics for metabolic disorders.

## 3. Gut Microbial Metabolites Derived from Dietary Fibers

Dietary fiber intake reduces the development of various diseases, such as inflammatory and metabolic disorders. Dietary fibers and indigestible oligosaccharides can escape host digestion and absorption into the small intestine [55]. However, anaerobic bacteria that reside in the gut can ferment these fibers into a wide variety of metabolites. Gut microbes produce SCFAs derived from dietary fibers as the final products of their metabolic process to maintain redox equivalence in the anaerobic environment of the gut. In the cecum and colon, most of the produced SCFAs (~95%) are rapidly absorbed into the colonic enterocytes, and the remaining 5% are maintained in feces. SCFAs act as the key regulators of energy metabolism and could play a potentially promising therapeutic role in preventing or mitigating metabolic disorders such as obesity and T2DM [10,56].

The three most abundant SCFAs in the human colon are acetate, propionate, and butyrate, which are found at an approximately 60:20:20 (millimolar concentration) ratio, respectively [56]. Metagenomic analysis of the gut microbiota in obese mice and humans showed that the expression of genes involved in carbohydrate metabolism is predominant. Furthermore, the transplantation of gut microbiota from obese individuals into germ-free mice significantly increased adiposity in the recipient mice, which was opposite to that observed after the transplantation of gut microbiota from lean individuals. Moreover, in European and Chinese cohort studies, despite their ethnic and dietary differences, patients with type II diabetes showed lower concentrations of butyrate and a higher population of Clostridiales, the non-butyrate producing bacteria [57]. SCFAs have been shown to affect the host immune system and metabolism through multiple mechanisms, including the regulation of histone acetylation and methylation, and via GPCRs [10,58]. SCFAs are involved in intestinal immune homeostasis because of their role in regulating the polarization and induction of T lymphocytes by differentiating T lymphocytes into effector and regulatory T cells (Treg cells). The inhibition of histone deacetylases (HDACs) is also a regulator mechanism of SCFAs, being carried out by butyrate and propionate but not acetate [59]. Recent studies have demonstrated that the ability of butyrate to promote the induction of forkhead box P3 gene expression was strongly correlated with its HDAC inhibitory activity, suggesting that butyrate regulates the differentiation of colonic Treg cells through epigenetic modification [60,61]. 

SCFAs regulate host energy homeostasis by activating GPR41, GPR43, and Olfr78 (members of the free fatty acids receptor (FFAR) family) in the sympathetic nervous system, adipose tissues, pancreas, intestine, and other tissues. In enteroendocrine cells in the intestine, SCFAs can trigger the secretion of peptide YY (PYY) and glucagon-like peptide-1 (GLP-1) via GPR41, GPR43, and Olfr78 [29,30,31]. Additionally, SCFAs have been positively associated with increased plasma levels of PYY and GLP-1 in both humans and mice [30,32]. Moreover, GPR43 expressed in white adipose tissues suppresses insulin signaling in adipocytes, which inhibits fat accumulation in adipose tissues and promotes the metabolism of unincorporated lipids and glucose in other tissues [33]. Although there are conflicting reports that describe the physiological functions of GPR43 in white adipose tissue, GPR43-expressing white adipose tissues may exert physiological effects based on the energy conditions and nutritional status of the body and may thereby contribute to the maintenance of energy homeostasis. In fact, Hong et al. investigated to clarify the role of GPR43 in adipocytes and showed that the expression of GPR43 in white adipose tissues of obese mice was significantly higher than in lean mice. The treatment of SCFAs in 3T3-L1 adipocytes remarkably induced GPR43 mRNA expression and adipogenesis, whereas siRNA-mediated suppression of GPR43 mRNA inhibited adipogenesis. Additionally, the treatment of SCFAs suppressed isoproterenol-induced lipolysis in a dose-dependent manner in 3T3-L1 cells [34]. Moreover, Ge et al. examined that these effects are mediated by GPR43 by showing SCFAs-induced suppression of lipolysis and the release of glycerol by adipocytes isolated from wild-type mice in vitro. Furthermore, although the activation of GPR43 via intraperitoneal injection of SCFAs led to a rapid reduction in plasma fatty acid levels in vivo, this effect was eliminated in GPR43-deficient mice [35]. By contrast, GPR41-mediated sympathetic nervous system activation leads to the secretion of noradrenaline and increased energy expenditure by raising the heart rate and body temperature [36]. Briefly, SCFAs treatment increased energy expenditure and heart rate in adult wild-type mice, which were not observed in GPR41-deficient mice. Since the effect of SCFAs on heart rate was attenuated by pretreatment with a β-adrenergic receptor blocker, propionate activates the sympathetic nervous system via GPR41 [36]. These results indicate that the biological functions of FFAR3, such as SNS activation by propionate, may be implicated in recognizing energy conditions in the body and may thereby contribute to the maintenance of energy homeostasis. These SCFA receptors, GPR41 and GPR43, play an important role in immune and metabolic responses, and SCFAs may mediate the crosstalk between gut microbiota and GPR41- and GPR43-expressing cells (Figure 2).

Because gut microbial metabolites including SCFAs translocate into the bloodstream and affect systemic regulation, SCFAs produced by maternal gut microbes also likely influence fetal development in utero. The Developmental Origin of Health and Disease (DOHaD) theory is based on the concept that the development of metabolic disorders is caused by genes and environmental factors, (e.g., nutrition, stress, or environmental chemicals) during the fetal period. The mechanisms linking maternal metabolic status during pregnancy to cardiometabolic disease risk in offspring remain elusive. Since the perinatal environment changes dramatically before and after birth, the fetus is abruptly forced to adapt to the new environment, which changes significantly even in physiological conditions [37]. Recent studies have demonstrated the role of maternal SCFAs on fetal development and postnatal growth. During pregnancy, GPR41 and GPR43 expressed in the sympathetic nerve, intestinal tract, and pancreas of the embryo are activated by maternal SCFAs and facilitate the development and shaping of the embryonic energy metabolism. Thus, the gut microbiota of pregnant mice provides an environmental signal that refines energy homeostasis in the offspring to prevent the development of metabolic disorders [38] (Figure 2) (Table 1).

## 4. Gut Microbial Metabolites Derived from Dietary Lipids

Since the composition and function of the gut microbiota are affected by dietary lipids, dietary lipids in turn may influence the host physiology through interaction with the gut microbes. Dietary lipids affect the gut microbiota both as metabolic substrates and antibacterial agents [62]. In the past decade, several studies have explored the role of specific dietary lipids in gut microbiota dysbiosis in relation to risk factors of metabolic disorders. Although saturated fatty acids are known to promote insulin resistance via Toll-like receptor and Jun N-terminal kinase signaling, their direct effects on host metabolism are less pronounced than the influence of the gut microbiota [63,64]. In fact, Bäckhed et al. reported that germ-free mice are protected against insulin resistance and obesity induced by a high-fat diet (HFD) rich in saturated fatty acids [65]. As demonstrated in mice, a diet containing high levels of saturated fatty acids or ω6 polyunsaturated fatty acids (PUFAs) induced obesity, whereas that containing high levels of ω3 PUFAs had anti-obesity effects. Although the feeding of these different dietary lipids affected the composition of gut microbiota in both groups of mice, the diversity and abundance of *A. muciniphila*, *Lactobacillus* and *Bifidobacterium* were found to be particularly higher in ω3 PUFAs rich diet-fed mice [20].

In response to dietary lipids, the gut microbiota can modulate host energy homeostasis through the production of bioactive metabolites that cannot be synthesized by mammalian hosts. Although most diet-derived LCFAs are absorbed into the small intestine, a few LCFAs will reach the colon and may therefore directly modulate the composition of the gut microbiota. The gut microbiota mediates carbon chain length, saturation, and double bond position of PUFAs derived from dietary lipids as a detoxifying mechanism in the gut [66]. For example, gut microbial processing of linoleic acid has been shown to produce metabolites, (e.g., conjugated linoleic acid; CLA) that may influence the physiology and health of the host. The CLA isomer, c9, t11-CLA, improves insulin sensitivity and decreases atherosclerosis by activation of proliferator-activated receptor γ (PPARγ) [67]. Additionally, LCFAs regulate host energy homeostasis via GPR40 and GPR120, which belong to the FFAR family, in the adipose tissues, pancreas, and intestine. The activation of GPR40 and GPR120 by LCFAs in enteroendocrine cells has been shown to promote the secretion of gut hormones, (e.g., GLP-1 and gastric inhibitory polypeptide) in both in vitro and in vivo models. In fact, Edfalk showed that the acute oral administration of fat stimulates both GLP-1 and GIP secretion in wild-type mice and that such stimulation does not occur in GPR40-deficient mice; therefore, GPR40-deficient mice treated with fat acutely exhibited low insulin levels and high glucose levels [68]. Additionally, Sundström et al. indicated that the activation of GPR120 led to decreased plasma glucose levels and increased insulin and GLP-1 secretion after intravenous administration of glucose in wild-type mice, but not in GPR120-deficient mice. [69]. Additionally, a GLP-1 receptor antagonist, exendin 9–39, suppressed the effects of GPR120 activation on plasma glucose levels and insulin secretion in wild-type mice [69]. In the pancreas, GPR40 is also expressed in pancreatic β cells, where its stimulation by LCFAs enhances insulin secretion. Steneberg et al. showed that the chronic effects of LCFAs on glucose-stimulated insulin secretion (GSIS) are reduced in the GPR40-deficient mice, whereas pancreatic β cell-specific GPR40 overexpression prevented the development of hyperglycemia in obese mice [70]. Additionally, Nagasumi et al. clarified that pancreatic β cell-specific GPR40 overexpression in KK mice, a model of obesity-associated diabetes, showed an improvement in insulin secretion and glucose tolerance [71]. Hence, the activation of GPR40 and GPR120 enhances GSIS not only by direct stimulation but also indirectly via the secretion of gut hormones [72,73]. Moreover, Ichimura et al. have shown that GPR120-deficient mice exhibit an obesity phenotype with decreased adipocyte differentiation and lipogenesis, suggesting a key role for GPR120 in lipid sensing and energy homeostasis [74]. These results provide insight into the contribution of LCFAs—which are often proposed as dietary supplements—to energy homeostasis via GPR40 and GPR120.

To date, various gut microbes have been identified as producing PUFA-derived intermediate metabolites that are not only less toxic but also beneficial [75]. These metabolites include several hydroxy and oxo fatty acids that affect processes related to host homeostasis. Indeed, in a mouse model of colitis, 10-hydroxy-*cis*-12-octadecenoic acid (HYA) was shown to restore intestinal epithelial barrier impairments by regulating tumor necrosis factor receptor 2 expression [76]. Additionally, the acute oral administration of HYA enhances GLP-1 secretion through GPR40 and GPR120 pathways and improved glucose homeostasis in mice. Interestingly, while HFD feeding altered the gut microbial composition and inhibited the microbial production of these metabolites, some gut microbes—through the production of PUFA metabolites—conferred to the host the ability to resist HFD-induced obesity (Figure 1) [39,50]. Similarly, another intermediate metabolite, 10-oxo-*trans*-12-octadecenoic acid (KetoA), can activate transient receptor potential vanilloid 1 and ameliorate adiposity and obesity-associated metabolic disorders [40]. Additionally, KetoA can activate PPARγ, induce adipocyte differentiation, and increase adiponectin production and glucose uptake in 3T3-L1 murine preadipocytes [41]. Moreover, KetoA has been shown to reduce both triglyceride accumulation in HepG2 hepatocytes by decreasing the expression of lipogenesis-related genes (*Srebp-1c*, *Scd-1*, and *Acc2*) and all risk factors related to cardiovascular diseases in the liver of mice fed a high-sucrose diet [42]. In summary, several studies have indicated that dietary PUFAs can be a substrate for bacteria of *Lactobacillus*, which metabolize the dietary PUFAs into intermediate metabolites, (e.g., HYA and KetoA) with different health benefits, including energy homeostasis. A clearer understanding of the metabolic processes regulated by the gut microbiota–as the basis of their healthy composition and interactions with the host–should be the new insight in the field of diet–gut microbiota research (Table 1).

## 5. Discussion

The impact of various nutrients that modulate the composition of the gut microbiota, or which are metabolized by the gut microbes into bioactive metabolites, is largely clarified in animal models and human trials. The induced changes of the gut microbes and their metabolites also contribute to health effects against metabolic disorders. This review has highlighted the contribution of the gut microbiota and its metabolites in host energy homeostasis as well as the effects of diets on metabolic disorders. Thus, the gut microbiota and its diet-derived metabolites provide an environmental cue that regulates host energy homeostasis to prevent the development of metabolic syndrome. These findings not only reveal the central mechanism underlying the role of diet in the interplay between the gut microbiota and host energy homeostasis but also potentially contribute to the development of functional foods for the prevention of metabolic disorders, (e.g., obesity and T2DM) through the tailored use of gut microbial metabolites. Additionally, the identification of the high levels of bioactivity resulting from the crosstalk between diet and gut microbiota via the microbial metabolites could promote the development of novel therapeutics for several diseases.

## Figures and Tables

**Figure 1 ijms-23-05562-f001:**
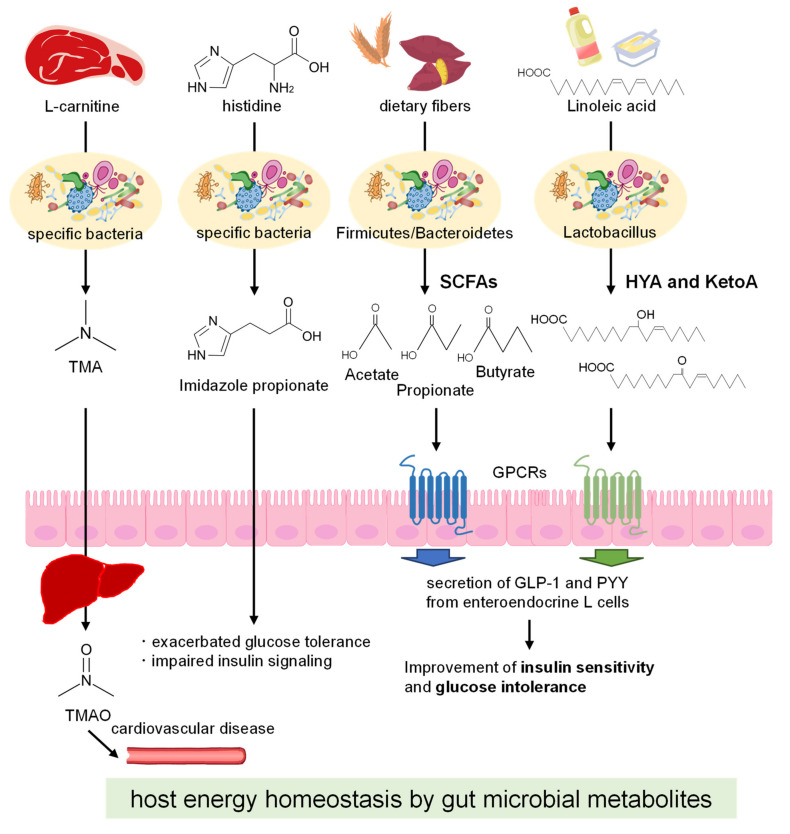
The diet-derived microbial metabolites in the host energy homeostasis.

**Figure 2 ijms-23-05562-f002:**
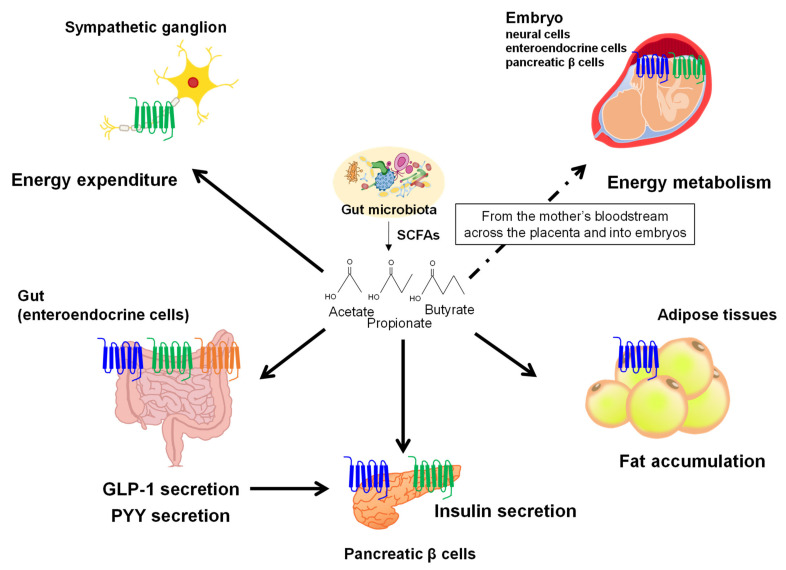
SCFAs, gut microbial metabolites derived from dietary fibers, regulate host energy homeostasis via GPCRs.

**Table 1 ijms-23-05562-t001:** Metabolic disorders-related gut microbiota and metabolites.

Gut Microbiota		Diseases	Ref.
Firmicute/Bacteroidetes ratio	increased	obesity	[12,13]
*Prevotella copri* and *Bacteroides vulgatus* (branched-chain amino acid-producing bacteria)	increased	obesity, type II diabetes	[19]
*Akkermansia muciniphila*	decreased	obesity	[16,17]
*Lactobacillus* and *Bifidobacterium*	decreased	obesity	[20]
Gut microbial metabolites			
trimethylamine N-oxide (TMAO)	increased	cardiovascular disease	[21,22]
branched-chain amino acids (BCAAs)	increased	obesity, type II diabetes	[19]
imidazole propionate	increased	type II diabetes	[23,24]
tauro-β-muricholic acid	increased	obesity	[25,26,27,28]
short-chain fatty acids (SCFAs)	decreased	obesity	[29,30,31,32,33,34,35,36,37,38]
10-hydroxy-*cis*-12-octadecenoic acid (HYA)	decreased	obesity	[39]
10-oxo-*trans*-12-octadecenoic acid (KetoA)	decreased	obesity, cardiovascular diseases	[39,40,41,42]

## Data Availability

Not applicable.

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
