# Peer review of "Involvement of Gut Microbial Metabolites Derived from Diet on Host Energy Homeostasis"

_ijms, 2022, doi:10.3390/ijms23105562_

Round 1

Reviewer 1 Report

This paper mainly introduces the impact of gut microbes on the host, especially on type 2 diabetes mellitus and obesity. Diet-derived metabolites provide environmental cues to modulate host energy homeostasis to prevent metabolic syndrome development. Free fatty acids have a great impact on metabolism in the body, however, this article discusses the Improvement of insulin sensitivity and glucose intolerance affected by GPCR from free fatty acids and different food sources. In addition, this article also introduces the information pathway is also an important key for gut microbes to affect cells and thus affect metabolism.

Specific comments

  1. This article provides a good background for the content, and even a shallow focus on drug development. In the text, the information pathways that gut microbes may affect are mentioned many times, which in turn affect insulin release and glucose balance. I hope the author can organize a new figure to propose the cellular information pathways and results of gut microbes' influence.
  2. Between paragraphs, it is suggested that authors can add subsections for easier reading, for example, 2. Gut microbiota and obesity can be subdivided into 2.1 Obesity-related information paths, etc.

Author Response

Reviewers1

This paper mainly introduces the impact of gut microbes on the host, especially on type 2 diabetes mellitus and obesity. Diet-derived metabolites provide environmental cues to modulate host energy homeostasis to prevent metabolic syndrome development. Free fatty acids have a great impact on metabolism in the body, however, this article discusses the Improvement of insulin sensitivity and glucose intolerance affected by GPCR from free fatty acids and different food sources. In addition, this article also introduces the information pathway is also an important key for gut microbes to affect cells and thus affect metabolism.

Thank you very much for your consideration and important suggestions. We have revised the manuscript based on these helpful comments and suggestions, and believe that the revised manuscript has been strengthened by the additional sentences and figures/tables, although the whole framework has not been changed fundamentally. Our point-by-point responses are as follows. Please see the attachment.

Specific comments

  1. This article provides a good background for the content, and even a shallow focus on drug development. In the text, the information pathways that gut microbes may affect are mentioned many times, which in turn affect insulin release and glucose balance. I hope the author can organize a new figure to propose the cellular information pathways and results of gut microbes' influence.

Thank you for your suggestion and we added the new figure 2.

  1. Between paragraphs, it is suggested that authors can add subsections for easier reading, for example, 2. Gut microbiota and obesity can be subdivided into 2.1 Obesity-related information paths, etc.

We changed as below,

2.1. Gut microbiota

2.2. Gut microbial metabolites

We believe that these corrections and revisions will be satisfactory and we hope that the revised manuscript will be acceptable for publication in the Journal.

Reviewer 2 Report

Reviewers

General comment-This is a clearly presented and well-written review by Miyamoto’s group. This review is focused on role of gut microbial metabolites derived from diet on host energy homeostasis. Dysfunction of energy balance leads to metabolic disorders such as diabetes, metabolic syndrome etc. These disorders are associated with changes in gut microbiome. Gut microbiota is affected by diets and nutrients and gut derived metabolites regulates host energy homeostasis. This review demonstrate the relationship between gut-derived metabolites and host energy homeostasis.

The proposed study is very interesting, results are explained well but I have the following comments and concerns.

  1. Study describes the interactions of gut microbial metabolites from dietary lipids, dietary fibers in host energy homeostasis. However, the studies cited are explained in descriptive manner. It will be great if you can show these in tabulated forms as well.

  1. Its known that gut microbiome regulates obesity. However, the mechanism of regulation is not explained well. If possible, please explain the possible modes of regulation by gut microbial metabolites on obesity.

  1. Gut microbial metabolites regulates host energy homeostasis however, it’s not clear whether gut microbial micronutrients (iron/copper essential cofactor for electron transport chain/oxidative phosphorylation) can regulate host energy homeostasis. Any explanation/ reports.

I recommend the manuscript be accepted for publication, with addressing these concerns.   

Author Response

Reviewers2

General comment-This is a clearly presented and well-written review by Miyamoto’s group. This review is focused on role of gut microbial metabolites derived from diet on host energy homeostasis. Dysfunction of energy balance leads to metabolic disorders such as diabetes, metabolic syndrome etc. These disorders are associated with changes in gut microbiome. Gut microbiota is affected by diets and nutrients and gut derived metabolites regulates host energy homeostasis. This review demonstrate the relationship between gut-derived metabolites and host energy homeostasis.

The proposed study is very interesting, results are explained well but I have the following comments and concerns.

Thank you very much for your kind consideration and various important suggestions. We have revised the manuscript based on the helpful comments and suggestions. Our point-by-point responses are as follows. Please see the attachment.

  1. Study describes the interactions of gut microbial metabolites from dietary lipids, dietary fibers in host energy homeostasis. However, the studies cited are explained in descriptive manner. It will be great if you can show these in tabulated forms as well.

Thank you for your suggestion.

We added the new table 1.

  1. Its known that gut microbiome regulates obesity. However, the mechanism of regulation is not explained well. If possible, please explain the possible modes of regulation by gut microbial metabolites on obesity.

We described about your comments in figure 2, which is focused on SCFAs and GPCRs

  1. Gut microbial metabolites regulates host energy homeostasis however, it’s not clear whether gut microbial micronutrients (iron/copper essential cofactor for electron transport chain/oxidative phosphorylation) can regulate host energy homeostasis. Any explanation/ reports.

Thank you for your beneficial comments. Although the micronutrients may also regulate the gut microbial components, it does not clarify in detail including molecular mechanisms and tissue- and cell-types. We will need to explain it in the other manuscript in the future.

I recommend the manuscript be accepted for publication, with addressing these concerns.

We believe that these corrections and revisions will be satisfactory and we hope that the revised manuscript will be acceptable for publication in the Journal.
